# Uncertainty Quantification of Knowledge Graph Embedding with Statistical Guarantees

**Yuqicheng Zhu[1,2], Nico Potyka[3], Daniel Hernández[1], Jiarong Pan[4], Bo Xiong[5], Yunjie He[1,2], Yuan He[6], Zifeng Ding[7], Evgeny Kharlamov[2,8], Steffen Staab[1,9]**

[1]*University of Stuttgart,* [2]*Bosch Center for AI,* [3]*Cardiff University,* [4]*Eindhoven University of Technology,* [5]*Stanford University,* [6]*University of Oxford,* [7]*University of Cambridge,* [8]*University of Oslo,* [9]*University of Southampton*

**Editors:** Leilani H. Gilpin, Eleonora Giunchiglia, Pascal Hitzler, and Emile van Krieken

## 1. Motivation

Knowledge Graph Embeddings (KGE) map entities and predicates into numerical vectors, providing reasoning capabilities by exploiting similarities and analogies between entities and relations (Bordes et al., 2011; Nickel et al., 2011; Sun et al., 2019). While KGE has been widely adopted in tasks such as link prediction and query answering, a critical limitation remains: the predictions lack reliable **uncertainty estimates** (Tabacof and Costabello, 2020; Safavi et al., 2020). This poses significant risks in high-stakes applications such as medical decision support, where reliable predictions and risk assessment are essential.

Consider a medical diagnosis query such as "*What diseases might Alice have?*". A typical KGE-based system returns a ranked list of candidate answers based on plausibility scores. However, these scores lack a probabilistic interpretation. For example, ranking common cold above lung cancer may mean: (1) the common cold is very likely while lung cancer is improbable, (2) both are likely but the common cold is slightly more probable, or (3) neither is likely, but the cold is relatively more plausible. These vastly different interpretations are indistinguishable from the ranking alone, making it unsuitable for contexts where risk-aware decisions are essential. For instance, a medical practitioner must know not just which diagnoses are more plausible, but also whether lung cancer can be confidently excluded.

Furthermore, recent work (Zhu et al., 2024) shows that KGE models can exhibit substantial variance in their predictions, even when achieving similar overall accuracy. This phenomenon, known as predictive multiplicity, stems from the non-convex optimization landscape of KGE training objectives, leading to models that capture different patterns and generalize inconsistently. As a result, individual predictions may vary significantly across models, highlighting the need for uncertainty quantification to support reliable decision-making in high-stakes domains.

## 2. Limitations of the Existing Work

Existing methods commonly rely on off-the-shelf calibration techniques, such as Platt scaling (Platt et al., 1999) and isotonic regression (Kruskal, 1964), to transform uncalibrated plausibility scores into well-calibrated estimates of prediction correctness. These techniques typically minimize the negative log-likelihood on a held-out calibration set (Tabacof and

Costabello, 2020; Safavi et al., 2020). However, they require access to ground-truth negative examples, which are generally unavailable in knowledge graphs (KGs). Negative triples are synthetically generated from positive ones, introducing bias and limiting the reliability of the calibration. As a result, these methods lack formal probabilistic guarantees and are highly sensitive to the choice of negative sampling strategy used during calibration.

## 3. KGCP and CondKGCP

To address this issue, our NAACL 2025 paper introduces KGCP (Zhu et al., 2025b), the first framework to reliably quantify uncertainty of KGE-based methods with statistical guarantees. Specifically, Given a query $q$ (e.g., $\langle Alice, hasDisease, ?\rangle$) and a user-specified confidence level $\alpha$ (e.g., 95%), KGCP outputs a set of candidate answers $C(q)$ such that the true answer $c$ is included with probability approximately $\alpha$:

$$\mathbb{P}(c \in C(q)) \approx \alpha. \tag{1}$$

The resulting answer set indicates how many candidate answers must be included to achieve the desired confidence level; thus, its size reflects the model's uncertainty for the given query.

However, KGCP only ensures marginal coverage—coverage is guaranteed on average over all queries. For many cases, especially where certain predicates represent more critical or less frequent relations, predicate-conditional guarantees are needed. To this end, our follow-up ACL 2025 paper proposes CondKGCP (Zhu et al., 2025a), a novel method that approximates predicate-conditional coverage guarantees.

$$\mathbb{P}(c \in C(q) \mid \mathrm{Pred}(q) = r) \approx \alpha, \tag{2}$$

where $\mathrm{Pred}(q)$ denotes the predicate of the query $q$, and $r$ is a specific predicate. The main challenges stem from the highly imbalanced distribution of triples across predicates in KGs. CondKGCP tackles this by (1) merging predicates with similar vector representations to enable reliable calibration in low-data regimes, and (2) using a dual calibration schema that combines both plausibility scores and entity ranks to construct compact answer sets.

## 4. Key Results and Contributions

- KGCP and CondKGCP are the first approaches to quantify uncertainty in KGE-based methods with formal *statistical guarantees*.

- We theoretically establish the validity of the coverage guarantees (Equations (1) and (2)) under mild and realistic assumptions. These guarantees are empirically validated across standard benchmark datasets and six representative KGE methods.

- The constructed answer sets are *tight*, meaning they provide concise and informative uncertainty estimates. For example, on WN18, KGCP yields answer sets with an average size of fewer than three entities, enabling effective decision support without overwhelming ambiguity.

- The answer sets are *adaptive* to query difficulty: harder queries lead to larger sets, reflecting higher uncertainty. This adaptiveness enables reliable, query-specific uncertainty estimation, which is essential for decision-making in high-stakes applications.

## Acknowledgments

The authors thank the International Max Planck Research School for Intelligent Systems (IMPRS-IS) for supporting Yuqicheng Zhu. The work was partially supported by EU Projects Graph Massivizer (GA 101093202), enRichMyData (GA 101070284) and SMARTY (GA 101140087), as well as the Deutsche Forschungsgemeinschaft (DFG, German Research Foundation) – SFB 1574 – 471687386. Zifeng Ding receives funding from the European Research Council (ERC) under the European Union's Horizon 2020 Research and Innovation programme grant AVeriTeC (Grant agreement No. 865958).

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
