# OpenReview forum: "Uncertainty Quantification of Knowledge Graph Embedding with Statistical Guarantees"
_nesyconf.org/NeSy/2025/Conference_Phase_2 — NeSy 2025 - Phase 2 Poster_

### Official Review · Reviewer_P59U · 2025-06-30
**Extended abstract: KGCP and CondKGCP**

**Rating:** 4
**Confidence:** 4

**Review:**

The extended asbtract describes two contributions: KGCP (NAACL 2025) and CondKGCP (ACL 2025).
These discuss probabilistic guarantees for queries on systems based on knowledge graph embeddings. As such, this is a valid topic for NeSy, and they seem like valuable contributions.

However, the extended abstract itself is not particularly good. I appreciate that the authors work with limited space, but the entire first page is spent on motivation and existing work. This results in far too little space left for the original contributions. The resulting explanation to me is inadequate. Also, the section 3 talks about UnKGCP, which is not referenced further in the text. Is this a type and meant to be CondKGCP?

**Anonymity:**

Remain anonymous

---

### Official Review · Reviewer_rit3 · 2025-07-06
**Interesting work but very vague even for an extended abstract**

**Rating:** 6
**Confidence:** 4

**Review:**

#### Summary
The paper discusses two recent contributions in developing knowledge graph embeddings that incorporate uncertainty. Specifically, the authors first summarise KGCP, a framework that, given a query, returns a set of answers that includes the true answer with a certain probability. Then, they summarise UnKGCP, which extends the same statistical guarantee to conditional queries.
#### Strengths
The setting of the work is interesting and highly relevant for the conference. Knowledge graphs currently encompass a large part of our organised symbolic knowledge and dealing with uncertainty on KGs is something that deserves more attention.

It seems the proposed two methods have theoretical guarantees and empirical evidence that the given answers, under some assumptions, include the true answer with a certain probability.
#### Weaknesses
While I understand that the format of an extended abstract is quite limiting, I would have preferred some more clarity on certain topics. Please find my comments and suggestions below.

**Some aspects could be made more precise.** You discuss in the related work how existing methods rely on certain calibration techniques to turn plausibility scores into probabilities, but then you never even give intuition on how you transform these scores into probabilities. This makes it very hard to asses, even intuitively, what your guarantees in Equation 1 and 2 are actually saying. Instead of vague statements like "under mild and realistic assumptions" (Section 4, point 2), just state those assumptions. In particular, state your distributional assumptions. In general, the authors do not give any intuition on *how*  they actually achieve their guarantees, which makes it hard to assess the impact of the claims.

**Why discuss both KGCP and UnKGCP?** From the discussion in Section 3, it seems that UnKGCP actually supersedes KGCP because it is able to deal with conditional queries. Hence, the discussion on KGCP seems redundant. Instead of discussion KGCP, why not go directly to UnKGCP? The reason I suggest this, is because that could free up the space needed to clarify some of the vagueness mentioned earlier.

**Some related works might be missing.** There are some other works that attempt probabilistic knowledge graph embeddings from a quite different point of view that are not mentioned [1, 2, 3]. Some of these methods model embeddings as random variables and apply probabilistic logic, e.g. PSL or ProbLog, to then deal with these distributions in a probabilistically sound fashion. It would be a good addition to see, again just intuitively, how the proposed method compares with those.

##### Smaller remarks
Is the name of the second method UnKGCP or CondKGCP? Both are used in the text. Specifically, UnKGCP is used in Section 3 while CondKGCP is mentioned in Section 4.
#### Verdict
The extended abstract is at times quite vague and having at least a high-level and intuitive understanding of what the proposed methods *actually do* would be quite valuable. Right now, the claims ring a bit hollow because of this vagueness. None the less, it discusses an interesting contribution that is highly topical for the NeSy conference.

[1] Chen, Xuelu, et al. "Embedding uncertain knowledge graphs." _Proceedings of the AAAI conference on artificial intelligence_. Vol. 33. No. 01. 2019.

[2] Vilnis, Luke, et al. "Probabilistic Embedding of Knowledge Graphs with Box Lattice Measures." _Proceedings of the 56th Annual Meeting of the Association for Computational Linguistics (Volume 1: Long Papers)_. 2018.

[3] Maene, Jaron, and Luc De Raedt. "Soft-unification in deep probabilistic logic." _Advances in Neural Information Processing Systems_ 36 (2023): 60804-60820.

**Anonymity:**

Disclose identity

---

### Official Review · Reviewer_vPLS · 2025-07-08
**Good paper, NeSy adjacent**

**Rating:** 7
**Confidence:** 4

**Review:**

**Summary**: The paper addresses the problem of probability calibration in knowledge graph embedding (KGE) research. The authors propose KGCP and CondKGCP, two model-agnostic frameworks to quantify uncertainty in KGE models with statistical guarantees. This can prove crucial in high-stakes applications, e.g. knowledge graph completion for biomedical purposes. For a given query, the proposed model provides answer sets with a certain confidence $\alpha$ that the true answer $c$ is included. The set sizes adapt to query difficulty, providing larger sets for harder queries, thus reflecting higher uncertainty.

**Quality**: The work is both novel and significant. While KGEs have been trained with probabilistic losses for many years (often BCE or CE losses), they are usually evaluated based on ranking objectives alone which, as the authors suggest, has some flaws. Recent works have investigated the calibration of probabilistic scores and showed that they are usually poorly calibrated (on top of the works already referred, I would also add [Loconte et al., 2023](https://arxiv.org/abs/2305.15944)). As far as I know this is the first work that focuses on conformal prediction sets for KGEs.

**Topicality**: Knowledge graphs are generally relevant to NeSy. They often act as a primary component for representing the symbolic knowledge that a neural system can access (e.g., for retrieval, reasoning, etc.). However, the way KGs are used in this paper, as in "a dataset for link prediction", is in my opinion more relevant to general ML than NeSy specifically. I find it *more NeSy-adjacent than NeSy-relevant*. Still, I think this is a good paper, and maybe it can inspire attendees to extend the method to other KGE-based tasks more directly relevant to NeSy.

**Anonymity:**

Remain anonymous